# Transcriptome and Metabolome Integrated Analysis of Two Ecotypes of *Tetrastigma hemsleyanum* Reveals Candidate Genes Involved in Chlorogenic Acid Accumulation

**DOI:** 10.3390/plants10071288

**Published:** 2021-06-24

**Authors:** Shuya Yin, Hairui Cui, Le Zhang, Jianli Yan, Lihua Qian, Songlin Ruan

**Affiliations:** 1College of Agriculture and Biotechnology, Zhejiang University, Hangzhou 310058, China; yin_shuya@126.com (S.Y.); hrcui@zju.edu.cn (H.C.); 2Institute of Biotechnology, Hangzhou Academy of Agricultural Sciences, Hangzhou 310058, China; sky6609@163.com; 3Institute of Crops, Hangzhou Academy of Agricultural Sciences, Hangzhou 310058, China

**Keywords:** *Tetrastigma hemsleyanum*, transcriptome, metabolome, chlorogenic acid

## Abstract

*T. hemsleyanum* plants with different geographical origins contain enormous genetic variability, which causes different composition and content of flavonoids. In this research, integrated analysis of transcriptome and metabolome were performed in two ecotypes of *T. hemsleyanum*. There were 5428 different expressed transcripts and 236 differentially accumulated metabolites, phenylpropane and flavonoid biosynthesis were most predominantly enriched. A regulatory network of 9 transcripts and 11 compounds up-regulated in RG was formed, and chlorogenic acid was a core component.

## 1. Introduction

*Tetrastigma hemsleyanum* Diels et Gilg is a herbaceous climber that is widely distributed in tropical to subtropical regions, mainly in provinces in south and southwest China, including Zhejiang, Jiangsu, Jiangxi and Tibet [1]. Its wild habitats are mainly hillside shrubs, valleys, and rock cracks under forests and by streams, at an altitude of 300–1300 m [2]. *T. hemsleyanum* is a valuable traditional herb known as “San ye qing (SYQ)”, and it is peculiar to China; the whole plant has been widely used as a broad-spectrum antibiotic material against fever and inflammation. Modern pharmacological studies have shown that *T. Hemsleyanum* also has antioxidant, antivirus, antitumor, immunomodulatory and hypoglycemic effects [3].

It has been determined that the biological activities of *T. hemsleyanum* can be attributed to multiple components, such as polysaccharides, polyphenols, terpenoids and alkaloids [4,5,6], and flavonoids are the most dominant and important components in *T. Hemsleyanum* [7]. Phenylalanine and tyrosine are transformed into precursors of flavonoids such as p-Coumaroyl-CoA and caffeoyl-CoA through the phenylpropanoid biosynthesis pathway, then these precursors enter the flavonoid biosynthesis pathway to form the basic flavonoid skeleton, which is modified to form different kinds of flavonoid products. [8].

Chlorogenic acid is generated as an intermediate product in the phenylpropanoid biosynthesis pathway. As a depside with antioxidant activity, chlorogenic acid is a widespread dietary component, with coffee and tea probably being the main dietary sources of chlorogenic acid [9]. Chlorogenic acid is also the main medicinal component of *Lonicera japonica* [10]. It has been found that chlorogenic acid exerts pharmacological effects in reducing incidence of several chronic and degenerative diseases [11].

Several studies have identified the key enzymes in basic flavonoids biosynthesis, which include phenylalanine ammonialyase (PAL), 4-coumarate-CoA ligase (4CL), trans-cinnamate 4-monooxygenase (C4H), chalcone synthase (CHS) and chalcone isomerase (CHI) in *T. hemsleyanum* [12,13], but the underlying mechanism of chlorogenic acid accumulation is still unclear in *T. hemsleyanum.*

*T. hemsleyanum* plants with different geographical origins contain enormous genetic variability, which causes different compositions and content of flavonoids [14,15]. This is convenient for ascertaining the regulatory networks of the flavonoid biosynthesis pathway. The integration of metabolomics and transcriptomics can provide precise information on gene-to-metabolite networks for identifying the function of unknown genes [16].

*T. hemsleyanum* ecotypes RG and SG13 were collected from entirely different primary environments, RG was formed in warm and moist regions with low diurnal variation, and SG13 was formed in regions with four distinct seasons that frequently experienced extreme climates. Among the ecotypes we collected, RG and SG13 belong to different clusters by phenotypic clustering (results not disclosed). Under cultivation conditions, RG and SG13 exhibited enormous differences in morphology (Appendix A), growing speed and stress resistance. Therefore, we explored the different pattern of chlorogenic acid and flavonoid biosynthesis in the *T. hemsleyanum* ecotypes RG and SG13 using integrated analysis of transcriptome and metabolome. The different expressed transcripts and differentially accumulated metabolites were most enriched in the phenylpropane and flavonoid biosynthesis pathway, especially the biosynthesis and accumulation of chlorogenic acid. We constructed a putative regulatory model and identified nine candidate genes involved in chlorogenic acid accumulation.

## 2. Results

### 2.1. Transcriptome Analysis in RG and SG13 Leaves

To explore the differences in gene expression between the two *T. hemsleyanum* cultivars, RNA was extracted from the third section of leaves from the stem tip for expression gene sequencing (Appendix A). After data filtering and assembly, a total of 55,373 unigenes were detected, of which 5428 were expressed differently in the two varieties, with 2406 and 3022 being expressed more in RG and SG13, respectively (Figure 1a). Principal component analysis (PCA) was employed to identify the differences in expression profiles among samples (Figure 1b). The RG and PL samples were separated by the first principal component (PC1, 43.56% of the total variables), indicating the expression patterns of unigenes were different in RG and SG13 leaves.

For functional annotation, KOBAS 2.0 software was used for KEGG enrichment analysis, 1029 (18.96%) different expressed genes were annotated to 119 KEGG reference pathways (Figure 2). The enrichment analysis was performed on DEGs expressed more in RG or SG13, respectively, DEGs participating in phenylpropane biosynthesis (ko00940) and flavonoid biosynthesis (ko00941) were predominantly enriched. Among 30 DEGs annotated to the phenylpropane biosynthetic pathway, 21 were more highly expressed in RG. Among 14 DEGs annotated to the flavonoid biosynthetic pathway, 10 were more highly expressed in RG. These results indicate the differences between two varieties of *T. hemsleyanum* in phenylpropane biosynthesis and the flavonoid biosynthesis model.

Additionally, DEGs up-regulated in RG were enriched in plant–pathogen interaction (ko04626), while DEGs up-regulated in SG13 were obviously enriched in plant hormone signal transduction (ko04075) and tyrosine metabolism (ko00350) pathway, indicating that there were more extensive differences between the two varieties.

### 2.2. Metabolic Differences in RG and SG13 Leaves

To further compare the differences in metabolites between the two varieties, the sample extracts were analyzed using an LC-ESI-MS/MS system. A total of 603 metabolites were detected in the two sets of samples, of which 236 were differentially accumulated metabolites. The contents of 131 metabolites were higher in RG, while those of 105 metabolites were greater in SG13 (Figure 3).

Differentially accumulated metabolites with high VIP values were mainly flavonoids, flavonols, quinate and its derivatives, such as quinic acid O-glucuronic acid, chlorogenic acid, and apigenin derivatives. The top 10 different metabolites by VIP value were all accumulated more in RG (Table 1), while the metabolites accumulated more in SG13 were mainly flavone glycosides.

KEGG enrichment analysis was also performed on differentially accumulated metabolites, phenylpropane biosynthesis and flavonoid related metabolites biosynthesis pathways had the best significance and rich factor. A total of 58 metabolites were mapped to flavonoid biosynthesis (ko00941), 54 were mapped to flavone and flavonol biosynthesis (ko00944), 14 were mapped to isoflavonoid biosynthesis (ko00943), and 19 metabolites were mapped to phenylpropanoid biosynthesis (ko00940). Additionally, differentially accumulated metabolites participating in purine, pyrimidine and tryptophan metabolism were also significantly enriched (Figure 4).

### 2.3. Integrated Analysis of Metabolites and Transcripts in the Phenylpropanoid and Flavonoid Biosynthesis Pathway

Integrated analysis indicated metabolites and transcripts participating in 49 KEGG reference pathways together, including flavonoid and phenylpropanoid biosynthesis, which revealed different metabolism patterns between RG and SG13. In the *T. hemsleyanum* cultivar RG, the expression of *ADT* (*arogenate/prephenate dehydratase)* was up-regulated, catalyzing L-Arogenic acid, the precursor of tyrosine and phenylalanine, to transform to phenylalanine. The expression of downstream genes in the phenylpropane synthesis pathway, namely six DEGs predicted as *PAL*, *4CL*, *C3H*, *CSE*, *HCT*, were all up-regulated, while chlorogenic acid and cinnamaldehyde accumulated more in RG. The expression of four DEGs predicted as *CCoAOMT*, *CHS* and *CHI* were up-regulated in the flavonoid biosynthesis pathway, and the content of naringenin chalcone and its derivatives increased remarkably, while at the same time, pinocembrin, butin and eriodictyol accumulated more in RG. The abundance of the metabolites and transcripts mentioned above is presented in Table 2, and the biosynthesis pathways are exhibited by Figure 5.

In SG13, the expression of the tryptophan biosynthesis gene *trpB (tryptophan synthase beta chain**)*, *TYRAAT* (*arogenate dehydrogenase*), which catalyzes L-Arogenic acidtransform to tyrosine, and *PTAL* (*phenylalanine/tyrosine ammonia-lyase*), which catalyzes tyrosine transform to coumaric acid for participating phenylpropanoid biosynthesis, were up-regulated. The expression of *DFR* (*bifunctional dihydroflavonol 4-reductase*), and the abundance of taxifolin and metabolites downstream, including catechin, L-epicatechin, myricetin, epigallocatechin and gallocatechol, increased.

Among the differential metabolites mentioned above, chlorogenic acid, pinocembrin and naringenin chalcone showed the highest abundance. The log2 fold change (FC) value of chlorogenic acid in RG compared to SG13 reached more than 21. To gain a better understanding of the regulatory network in RG, we performed a Pearson correlation analysis on the 10 genes and 11 metabolites involved in the chlorogenic acid and flavonoid biosynthesis pathways (Figure 6). The results showed that there were 89 significant correlation combinations between the genes and metabolites, with the Pearson correlation coefficient >0.8 and *p*-value < 0.05. A regulatory network based on significant correlation of nine genes and all metabolites were constructed, except that no significant correlations were found between *4CL* and any of the metabolites.

### 2.4. Validation of Expression of Genes Involved in Chlorogenic Acid Accumulation

Real-time quantitative PCR (qPCR) was used to verify the transcription of differential expressed genes involved in chlorogenic acid accumulation (Figure 7). Results showed that the expression levels of six genes in RG were higher than those in SG13, especially two possible *HCT* genes annotated to k13065 in KEGG analysis, including hydroxycinnamoyl-CoA quinate hydroxycinnamoyl transferase and hydroxycinnamoyl-CoA shikimate/quinate hydroxycinnamoyl transferase. The expression level of c100482.graph in RG was 100 times greater than that in SG13, and the expression level of c78409 was relatively low; furthermore, it was only expressed in RG, where it may be the key gene causing the difference of chlorogenic acid accumulation between RG and SG13.

## 3. Discussion

Flavonoids are important secondary metabolites in *T. hemsleyanum*, which was synthesized through phenylpropane synthesis pathway. In the present study, the differences in transcripts and metabolites in two different *T. hemsleyanum* varieties, RG and SG13, were compared. Transcriptome analysis results showed that 30 differentially expressed unigenes were assigned to the phenylpropanoid biosynthesis pathway. Most of the genes associated with chlorogenic acid accumulation in the phenylpropanoid biosynthesis pathway were upregulated in RG (Figure 5), indicating that the chlorogenic acid biosynthesis was promoted in RG. Results of metabolome analysis using ultra-performance liquid chromatography and tandem mass spectrometry supports this conclusion; higher abundance of chlorogenic acid was detected in RG, but there was no signal in SG13 (Table 1), and 1 *PAL*, 1 *4CL*, 1 *C3H*, 1 *CSE* and 2 *HCT* were up-regulated in RG, putatively causing the differential accumulation of chlorogenic acid.

Two transcripts annotated to KEGG position K13065 and predicted as *HCT* genes up-regulated most, namely *c100482.graph_c0*, which increased 33-fold, and *c78409.graph_c0*, which was only expressed in RG (Figure 5). Consistent with transcriptome data, quantitive real-time PCR demonstrated five transcripts associated with chlorogenic acid biosynthesis expressed more in RG, among them, expression of *c100482.graph_c0* increased over 100-fold (Figure 7).

CCoAOMT involved in chlorogenic acid metabolism by changing caffeoyl-CoA to feruloyl-CoA for flavonoid and lignin biosynthesis [17]. In this study, expression of *CCoAOMT* was also up-regulated in RG, and facilitated the accumulation of eriodictyol, the product of feruloyl-CoA. Accumulation of chlorogenic acid promoted the production of flavonoids in RG. Except for eriodictyol, accumulation of pinocembrin, butin, naringenin chalcone and its derivatives were up-regulated, which transformed from cinnamoyl-CoA or p-coumaroyl-CoA in chlorogenic acid biosynthesis pathway by CHS and modified by CHI [18,19] (Figure 5). In this study, 14 differentially expressed unigenes were assigned to the flavonoid biosynthetic pathway, 2 CHS and 1 CHI putatively caused the differential accumulation.

To gain a better understanding of the regulatory network in RG, 10 genes and 11 metabolites up-regulated in RG were identified, and 89 significant correlation associations were found. In addition to *4CL*, there is a general association between other genes and metabolites. Chlorogenic acid is one of the cores in the network (Figure 7).

Chlorogenic acid is an important intermediate product in the phenylpropane synthesis pathway, and at the same time, it is also a kind of dietary polyphenol with biological activity. It has many important therapeutic effects, such as antioxidant activity, antibacterial, hepatoprotective, cardioprotective, anti-inflammatory, antipyretic, neuroprotective, antiobesity, antiviral, anti-microbial, and anti-hypertension, and is a free radicals scavenger and a central nervous system stimulator [20]. In plants, chlorogenic acid acting as defence metabolite is probably linked to defense signaling networks activated by salicylic acid and methyl-jasmonate [21] and participate in combating pathogens, driving off predators, protecting cells from oxidative damage when exposed to salinity, heat and UV [22,23,24]. The accumulation of chlorogenic acid is regulated differently in different species; for example, *C4H* silence causes the accumulation of cinnamic acid accompanied by significant reductions in p-coumaric acid in sweet sagewort, but *C4H* mutation results in the accumulation of cinnamoylmalate in *Arabidopsis thalania* [11,25,26]. Regulatory genes that were highly correlated with the accumulation of chlorogenic acid were identified in this study, providing foundations for further research on chlorogenic acid in *T. hemsleyanum*.

## 4. Materials and Methods

### 4.1. Plant Materials and Growth Conditions

Two ecotypes of *Tetrastigma hemsleyanum* (Sanyeqing) grown in the plant garden of the Hangzhou Academy of Agricultural Sciences (Hangzhou, Zhejiang Province, China) were involved in this research. RG was collected from Guangxi, with medium-sized acuminatissima leaves, and SG13 was collected from Zhejiang with small acuminate leaves. Leaves at the third node away from the top were collected for RNA and metabolite isolation, as well as for quantitative real-time PCR analysis (Appendix A).

### 4.2. RNA Isolation and Transcriptome Sequencing

Each ecotype was analyzed using three biological replicates, and transcriptome sequencing was performed by BIOMARKER TECHNOLOGIES (Beijing, China).

Total RNA was extracted by RNAprep Pure Plant Kit (Tiangen, DP441, Beijing, China). RNA concentration was measured using NanoDrop 2000 (Thermo, Waltham, MA, USA). RNA integrity was assessed using the RNA Nano 6000 Assay Kit of the Agilent Bioanalyzer 2100 system (Agilent Technologies, Santa Clara, CA, USA). A total amount of 1 μg RNA per sample was used as input material for the RNA sample preparations.

The transcriptome sequence library was constructed using NEBNext Ultra RNA Library Prep Kits for Illumina (NEB, Ipswich, MA, USA) following the manufacturer’s recommendations and index codes were added to attribute sequences to each sample. Briefly, mRNA was purified from total RNA using poly T oligo-attached magnetic beads and cDNA was synthesized using random hexamer primer and M-MuLV Reverse Transcriptase. After adenylation of 3′ ends of DNA fragments, NEBNext Adaptor with hairpin loop structure were ligated to prepare for hybridization. To select cDNA fragments of preferentially 240 bp in length, the library fragments were purified with AMPure XP system (Beckman Coulter, Beverly, CA, USA). Then 3 μL USER Enzyme (NEB, USA) was used with size-selected, adaptor-ligated cDNA at 37 °C for 15 min followed by 5 min at 95 °C. Then PCR was performed with Phusion High-Fidelity DNA polymerase, Universal PCR primers and Index (X) Primer. Finally, PCR products were purified (AMPure XP system) and library quality was assessed on the Agilent Bioanalyzer 2100 system. The clustering of the index-coded samples was performed on a cBot Cluster Generation System using TruSeq PE Cluster Kit v3-cBot-HS (Illumia, San Diego, CA, USA).

Sequencing was performed on an Illumina HiSeq 2500 platform (Novogene, Hong Kong, China). Raw reads of fastq format were firstly processed through in-house perl scripts, clean reads were obtained. The left files and right files from all samples was accomplished using Trinity. Total number of reads per kilobase per million reads (RPKM) of each gene was calculated based on the length of the gene and the counts of reads mapped to this gene. Differential expression analysis of two ecotypes was performed using the DESeq R package (1.10.1). Genes with an adjusted *p*-value <0.05 found by DESeq were assigned as differentially expressed. GO annotation was implemented using Blast2GO software and KOBAS (2.0) software was used for KEGG enrichment analysis of differentially expressed genes.

The sequence data in this study have been deposited in NCBI SRA with the accession number SRR14575715-SRR14575720 under bioproject accession PRJNA730723 (https://dataview.ncbi.nlm.nih.gov/object/PRJN A730723?reviewer=andv05obnsa3b8gao2a2ok000f, accessed on 19 May 2021)

### 4.3. Metabolic Profiling

The freeze-dried leaf was crushed using a mixer mill (MM 400, Retsch) with a zirconia bead for 1.5 min at 30 Hz. An amount of 100 mg of powder was weighed and extracted overnight at 4 °C with 1.0 mL 70% aqueous methanol. Following centrifugation at 10, 000g for 10 min [27], the extracts were absorbed (CNWBOND Carbon-GCB SPE Cartridge, 250mg, 3ml; ANPEL, Shanghai, China, www.anpel.com.cn/cnw, accessed on 19 May 2021) and filtrated (SCAA-104, 0.22μm pore size; ANPEL, Shanghai, China, http://www.anpel.com.cn/, accessed on 19 May 2021) before LC-MS analysis. The sample extracts were analyzed using an LC-ESI-MS/MS system.

HPLC (Shim-pack UFLC SHIMADZU CBM30A system, www.shimadzu.com.cn/, accessed on 19 May 2021) analytical conditions were as follows: column was Waters ACQUITY UPLC HSS T3 C18 (1.8 µm, 2.1 mm × 100 mm); the solvent system was water (0.04% acetic acid): acetonitrile (0.04% acetic acid); the gradient program was 95:5 *v*/*v* at 0 min, 5:95 *v*/*v* at 11.0 min, 5:95 *v*/*v* at 12.0 min, 95:5 *v*/*v* at 12.1 min, 95:5 *v*/*v* at 15.0 min; the flow rate was 0.40 mL/min; temperature was 40 °C; injection volume 2 μL. 

The effluent was alternatively connected to an ESI-triple quadrupole-linear ion trap (Q TRAP)-MS (MS, Applied Biosystems 6500 Q TRAP, www.appliedbiosystems.com.cn/, accessed on 19 May 2021), ESI Turbo Ion-Spray interface, operating in a positive ion mode and controlled by Analyst 1.6.3 software (AB Sciex), parameters were as follows: ion source, turbo spray; source temperature 500 °C; ion spray voltage (IS) 5500 V; ion source gas I (GSI), gas II (GSII), curtain gas (CUR) were set at 55, 60, and 25.0 psi, respectively; the collision gas (CAD) was high. Then LIT and triple quadrupole (QQQ) scans were acquired, DP and CE for individual MRM transitions was done with further DP and CE optimization. A specific set of MRM transitions were monitored for each period according to the metabolites eluted within this period.

The mass fragmentations were compared to the HMDB (http://www.hmdb.ca, accessed on 19 May 2021), METLIN (http://metlin.scrippps.edu, accessed on 19 May 2021) and KEGG (http://kegg.jp, accessed on 19 May 2021) databases. Differentially metabolic compounds were defined as those fold change ≥ 2 or ≤0.5, and among them, with a variable importance for projection (VIP) value ≥ 1 between two accessions. The obtained data were used by SIMCA-P V12.0.0 Demo (Umetric, Umea, Sweden) for principal component analysis (PCA) and partial least-squares discriminant analysis (PLS-DA).

### 4.4. Verification of Candidate Genes by Quantitative Real-Time PCR (qRT-PCR)

RNA samples were prepared by RNeasy Plant Mini Kit (Qiagen, Hilden, Germany) and DNA were eliminate by RNase-Free DNase Set (Qiagen). cDNAs were synthesized using SuperScript™ III First-Strand Synthesis SuperMix for qRT-PCR (Invitrogen, Waltham, MA, USA). Samples were amplified in Power SYBR^®^ Green PCR Master Mix (Applied Biosystems) and detected by CFX384 Real-time PCR system (Bio-Rad, Hercules, CA, USA). Three samples were performed for each cultivar and three replicates performed for each sample *T. hemsleyanum EF1α* was used as the reference gene. The primers were designed using beacon designer 7.8 and Primer Premier 6.0. The sequences of genes and primers are provided in Appendix A.

## 5. Conclusions

We performed an integrated analysis of transcriptome and metabolome in *T. hemsleyanum* varieties with middle-sized leaves (RG) and small leaves (SG13). A regulatory network of differentially expressed metabolites and transcripts in phenylpropanoid and flavonoid biosynthesis was found. We identified some candidate genes involved in chlorogenic acid accumulation, including five genes most likely involved in controlling chlorogenic acid biosynthesis, one gene possibly involved in catalyzing chlorogenic acid to flavonoid biosynthesis, and three genes involved in controlling flavonoid biosynthesis, which will provide a potential platform for functional genomic research.

## Figures and Tables

**Figure 1 plants-10-01288-f001:**
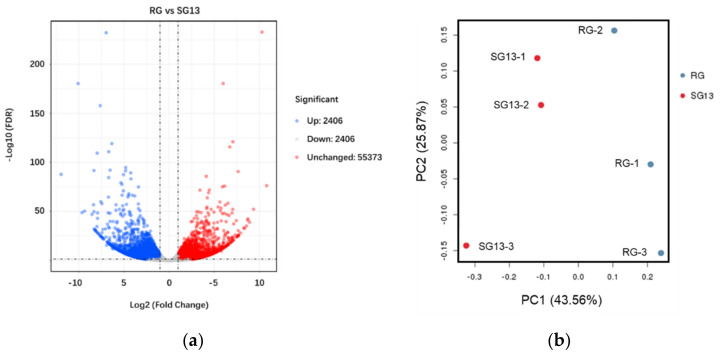
Difference in the expression patterns of two *T. hemsleyanum* ecotypes. (**a**) Volcano plot of differentially regulated genes in RG vs. SG13. The red dots were expressed more in SG13 and blue dots were expressed more in RG. (**b**) 2D PCA score plot of RG vs. SG13.

**Figure 2 plants-10-01288-f002:**
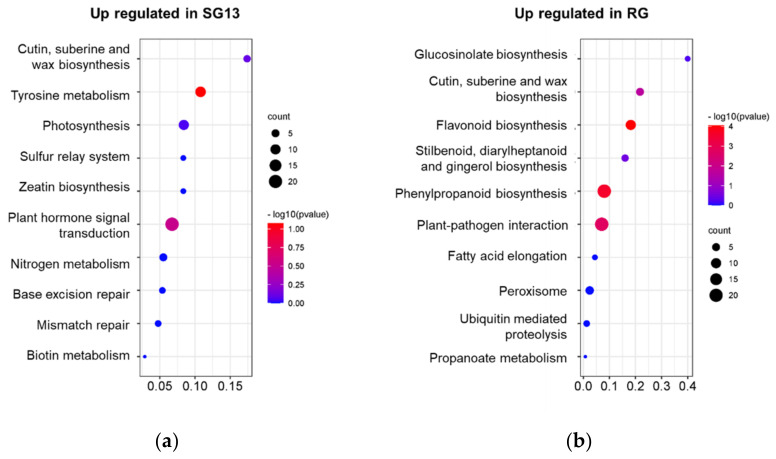
Statistics of KEGG pathway enrichment. The 10 most significant catalogues with lowest corrected *p*-value based on DEGs expressed more in (**a**) RG and (**b**) SG13, respectively.

**Figure 3 plants-10-01288-f003:**
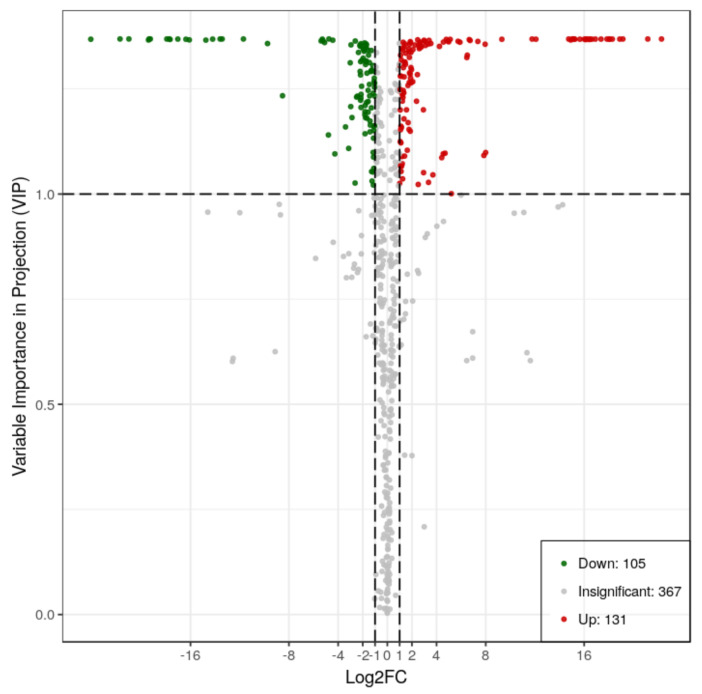
VIP (variable importance for projection) plot of differentially accumulated metabolites in RG vs. SG13 leaves. Red dots are metabolites accumulated more in RG, and green dots are metabolites accumulated more in SG13. Differentially accumulated metabolites were defined as those with a fold change ≥2 or ≤0.5, and variable importance for projection (VIP) value > 1.

**Figure 4 plants-10-01288-f004:**
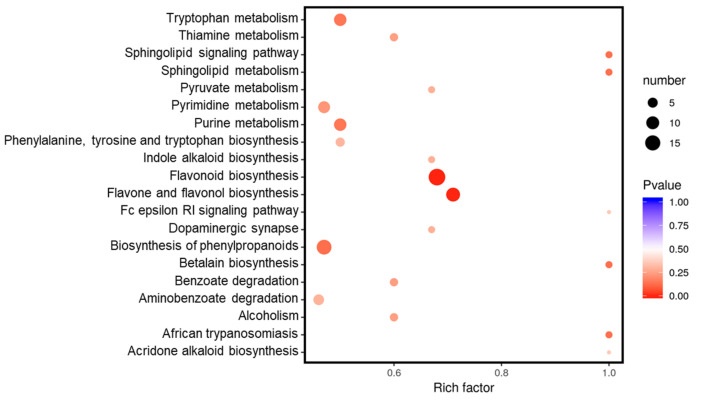
KEGG pathway enrichment of all differentially accumulated metabolites in RG vs. SG13 leaves. The 20 most significant catalogues with lowest corrected *p*-value are shown.

**Figure 5 plants-10-01288-f005:**
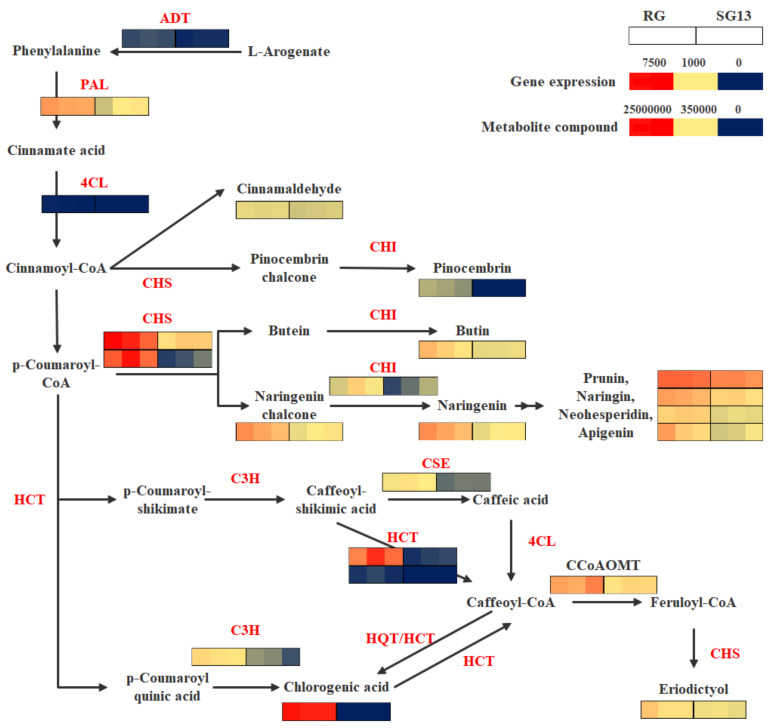
Phenylpropanoid and flavonoid biosynthesis upregulated in RG. This pathway is constructed based on the KEGG pathway. Each colored cell represents the normalized intensity of each compound according to the color scale (three biological replicates for each variety). Genes different expressed including prephenate dehydratase (ADT), phenylalanine ammonia-lyase (PAL), 4-coumarate-CoA ligase (4CL), 5-O-(4-coumaroyl)-D-quinate 3′-monooxygenase (C3H), caffeoylshikimate esterase (CSE), shikimate O-hydroxycinnamoyltransferase (HCT), caffeoyl-CoA O-methyltransferase (CCoAOMT), chalcone synthase (CHS) and chalcone isomerase (CHI).

**Figure 6 plants-10-01288-f006:**
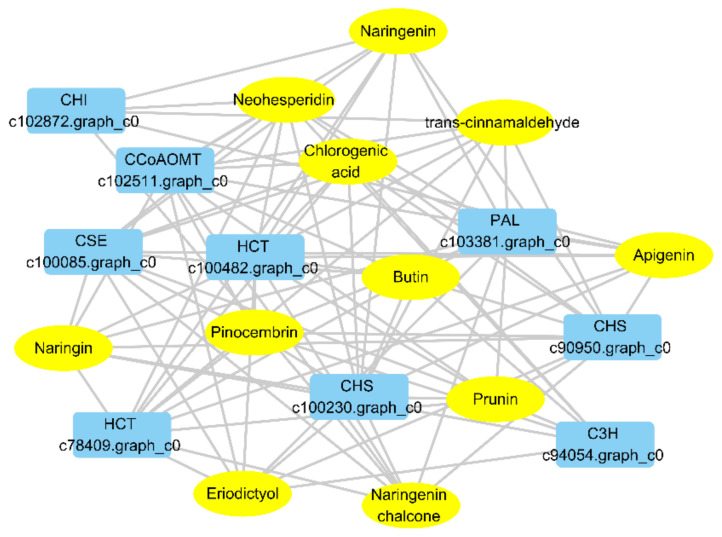
Connection network between differentially accumulated metabolites and related genes in RG. The networks between metabolites and transcripts were visualized with Cytoscape software (version 2.8.2).

**Figure 7 plants-10-01288-f007:**
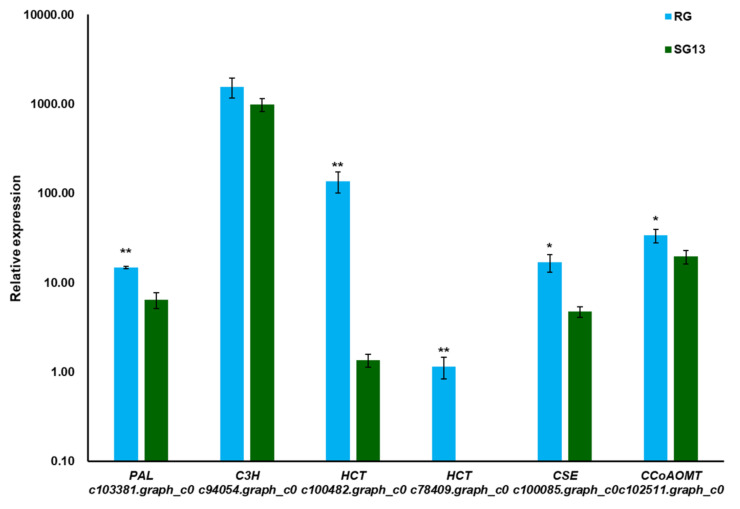
qPCR analysis of chlorogenate accumulation-related genes in RG and SG13. The gene expression was determined by fold change relative to the reference gene *EF1α*. Values are means of three biological repeats. Error bars indicate ± SE. Statistical significance was determined by *t*-tests, * and ** above the bar represent significant difference (*p* < 0.05) and extremely significant difference (*p* < 0.01), respectively.

**Table 1 plants-10-01288-t001:** List of top 10 metabolites by variable importance for projection (VIP) value.

Compounds	Class	RG	SG13	VIP	Log2 (Fold_Change)
Quinic acid O-glucuronic acid	Quinate and its derivatives	31,266,667	9	1.3688	−21.73
Chlorogenic acid (3-O-Caffeoylquinic acid)	Quinate and its derivatives	18,966,667	9	1.3688	−21.01
Isovitexin 7-O-glucoside (Saponarin)	Others	6,086,667	9	1.3688	−19.37
C-hexosyl-apigenin O-pentoside	Flavone C-glycosides	5,670,000	9	1.3688	−19.26
C-hexosyl-apigenin O-hexosyl-O-hexoside	Flavone C-glycosides	2,243,333	9	1.3687	−17.93
Kaempferol 3-O-rhamnoside (Kaempferin)	Flavonol	1,956,667	9	1.3687	−17.73
p-Coumaroyl quinic acid O-glucuronic acid	Quinate and its derivatives	9	300,000	1.3687	15.02
Chrysin 5-O-glucoside (Toringin)	Flavone	98,967	9	1.3688	−13.42
Tricin 4′-O-(syringyl alcohol) ether 5-O-hexoside	Flavonolignan	29,900	9	1.3687	−11.70
Acetyl tryptophan	Amino acid derivatives	16,166,667	601,000	1.3688	−4.75

Differentially accumulated metabolites were defined as those fold change ≥2 or ≤0.5, and variable importance for projection (VIP) value >1. Metabolite content were measured by average counts per second (CPS) of 3 repeated samples.

**Table 2 plants-10-01288-t002:** List of transcripts and metabolites up-regulated in RG.

	Accession Number	Function Prediction	KEGG ID	RG Count	SG13 Count	Fold Change (log2FC)
**Transcripts up-regulated in RG**	c96568.graph_c0	ADT	K05359	144	41	1.58
	c103381.graph_c0	PAL	K10775	2866	805	1.63
	c29476.graph_c0	4CL	K01904	12	0	3.08
	c94054.graph_c0	C3H	K09754	1166	350	1.46
	c100085.graph_c0	CSE	K18368	864	334	1.14
	**c100482.graph_c0**	**HCT**	**K13065**	**4779**	**118**	**5.04**
	**c78409.graph_c0**	**HCT**	**K13065**	**95**	**0**	**6.16**
	c102511.graph_c0	CCoAOMT	K00588	3104	1280	1.06
	c100230.graph_c0	CHS	K00660	6157	1532	1.80
	c90950.graph_c0	CHS	K00660	5452	225	4.32
	c102872.graph_c0	CHI	K01859	995	341	1.31
	**Index**	**Metabolite name**	**KEGG ID**	**RG Content**	**SG13 Content**	**Fold Change (Log2FC)**
**Metabolites up-regulated in RG**	**pme0398**	**Chlorogenic acid**	**C00852**	**18,966,667**	**9**	**21.00704**
	pme0424	trans-cinnamaldehyde	C00903	94,433	45,900	1.040802
	pme2979	Pinocembrin	C09827	7690	9	9.738843
	pme3473	Butin	C09614	516,000	114,000	2.178337
	pme2960	Naringenin chalcone	C06561	1,218,667	225,667	2.433038
	pme0377	Naringenin	C00509	1,182,333	209,133	2.499142
	pme0002	Neohesperidin	C09806	487,000	95,667	2.347834
	pme0330	Naringin	C09789	1,046,667	410,333	1.350934
	pme0371	Prunin	C09099	3,910,000	1,950,000	1.003694
	pme0379	Apigenin	C01477	750,000	102,367	2.873145

Expression and metabolite content were measured by average counts of samples in triplicate. Red words indicate the most up-regulated transcripts and metabolite.

## Data Availability

The data presented in this study are available within the article and its Appendix A.

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
