# Peer review of "Transcriptome and Metabolome Integrated Analysis of Two Ecotypes of Tetrastigma hemsleyanum Reveals Candidate Genes Involved in Chlorogenic Acid Accumulation"

_plants, 2021, doi:10.3390/plants10071288_

Round 1

Reviewer 1 Report

This is a review of the manuscript submitted by Yin et al. to Plants. In this paper, the authors compare transcriptomes and metabolomes of two ecotypes of Tetrastigma hemsleyanum.  

The idea of combining transcriptomics and metabolomics data was good, however, the weak side of this paper is a data processing and their visualization.  

Please find below points that should be improved.

 The last paragraph of the introduction should contain the main findings of the research.

Please explain somewhere in the text what was the criteria for choosing RG and SG13 T. hemsleyanum ecotypes.

The description of all Figures is unclear and does not meet the requirements of RNA sequencing data presentation.   

The quality of the Figures is very poor.

Figure 7. What type of statistics was performed? Please provide the long name for the reference gene.

Figure 5. Figure legend should explain what does mean ADT, PAL …. for function prediction.

Methods

How RNA was isolated for RNA-sequencing.

How many biological replicates were prepared?

How raw reads were trimmed from adaptors?

Information about the reference genome of T. hemsleyanum is missing.

What was the threshold for finding DEGs?

Presentation of RNA-seq data should start from PCA data to show a sample and biological replicates similarity.  

Supplementary data

RNA-seq Supplementary dataset is missing. This dataset should be presented as an excel file and should contain information for each gene – Gene ID, description of the gene, number of reads, RPKM for each biological replicate, mean RPKM +/- SD, fold- change, and adjust p-value.

The manuscript has lots of language errors.

For example, L: 98 “aredownregulated” should be “are down-regulated”

L:  193 Sentence cannot start with number ”2 transcripts annotated to KEGG position…”

Author Response

Dear reviewer:

Thank you very much for comments and suggestions.

We have finished our revision and tried our best to modified the problems mentioned in comments.

We provided the details of the revisions and responses to the comments blow.

Comments. The last paragraph of the introduction should contain the main findings of the research.

We have added main findings of the research in introduction.

Comments. Please explain somewhere in the text what was the criteria for choosing RG and SG13 T. hemsleyanum ecotypes.

We added the foundation we chose RG and SG13 in the last paragraph of the introduction.

Comments. Presentation of RNA-seq data should start from PCA data to show a sample and biological replicates similarity. 

We added the 2D PCA score plot as Figure 1. (b) and described it in line 74-78

Comments. The quality of the Figures is very poor. The description of all Figures is unclear and does not meet the requirements of RNA sequencing data presentation.

Considered transcriptome and metabolome both part of our results, we omitted different genes and metabolits that cannot be integrated, just exhibited and described our main main finding. We have constructed part of figures again to make it clear.

Comments. Figure 5. Figure legend should explain what does mean ADT, PAL …. for function prediction.

We have added the complete name involve in figure legend. KEGG ID of these genes we have provided in Table 2 in order that the function of related genes can be found easily in KEGG official website.

Comments. Figure 7. What type of statistics was performed? Please provide the long name for the reference gene.

Statistical significance was determined by T-tests. Reference gene we used was T. hemsleyanum elongation factor 1 alpha (EF1α). It is a unigene we found in another transcriptomic project. We have verified its expression is stable in 5 different T. hemsleyanum ecotypes and data of another RT-PCR used it as reference gene have been reported (DOI: 10.1371/JOURNAL.PONE. 0230154). We added its sequence in Table 1.

Comments. How RNA was isolated for RNA-sequencing.

Total RNA was extracted by column method. The plant tissue is rich in polysaccharides and polyphenolics, we used RNAprep Pure Plant Kit (Tiangen, DP441). We described it in 4.2, line 309.

Comments. How many biological replicates were prepared?

The experiment was performed using 3 biological replicates. We described it in 4.2, line 307.

Comments. How raw reads were trimmed from adaptors?

Before adaptor ligated, cDNA has been purified and size-selected. Then adaptor-ligated cDNA was amplified by PCR with Universal PCR primers and Index (X) Primer. The clustering was performed on PCR products by TruSeq PE Cluster Kit v3-cBot-HS (Illumia). After cluster generation, raw reads were generated by Illumina Hiseq 2000 platform. We described it in 4.2, line 314-335.

Comments. What was the threshold for finding DEGs?

Genes with an adjusted P-value <0.05 found by DESeq were assigned as differentially expressed. We described it in 341-343.

Comments. Information about the reference genome of T. hemsleyanum is missing.

没有关于T. hemsleyanum参考基因组的信息。

There is not genome sequencing of T. hemsleyanum been reported yet. We have performed genome sequencing now, but data registration has not been completed. In this research we analyzed transcriptome data without reference genome. After communicated with technician we found TopHat 2.0.9 were not used, so we have corrected it.

Comments. RNA-seq Supplementary dataset is missing. This dataset should be presented as an excel file and should contain information for each gene – Gene ID, description of the gene, number of reads, RPKM for each biological replicate, mean RPKM +/- SD, fold- change, and adjust p-value.

We added dataset of all unigenes different expressed between RG and SG13 as Table S2. As there is not genome of T. hemsleyanum been reported yet, we described unigenes as KRGG annotation and Nr annotation.

Comments. The manuscript has lots of language errors.

We checked and corrected errors we found.

Reviewer 2 Report

The work carried out by the authors and described in the manuscript seems to be scientifically relevant, showing the relationship between the gene expression and metabolite accumulation in two different cultivars of a plant species used in Tradidional Chinese Medicine. The results presented by the authors are very interesting and they can improve the knowledge regarding the genetic mechanism to the accumulation of the metabolites in this plant species, which ones are related to the biological properties of its extracts for medicinal purposes.

Despite the interesting results presented in the manuscript, I have one question I would like to do to the authors regarding some minor problem in the work.

In Page 9, Line 254 (Section 4.3. Metabolic profiling), the authors describe the procedure of metabolite extraction as follows: '100mg powder was weighted and extracted overnight at 4 ℃ with 1.0 ml 70% aqueous methanol.'; Why did the authors choose to perform an overnight extraction? Did they consider the possibility that, even at low temperature, the prolonged extraction time could have formed artifacts from reactions such as hydrolysis or conjugation between metabolites? The questioning is related to the fact that the extraction mixture consists of two polar protic solvents that can favor different reactions between organic molecules. 

The main reason for this question is because since the aim of the study is to identify a relationship between the gene expression and the phenylpropanoid metabolites (mainly the chlorogenic acid) accumulation in Tetrastigma hemsleyanum, it is essential, to ensure the reliability of the results, that the composition of metabolites analyzed and identified in extract samples be the more similiar as possible as those present in plant tissues.

I congratulate the authors for the work they carried out and the question presented above is the only point of the work for which I would like a more detailed explanation.

Author Response

Dear reviewer:

Thank you very much for comments and suggestions.

You asked “even at low temperature, the prolonged extraction time could have formed artifacts from reactions such as hydrolysis or conjugation between metabolites”. We got this extraction from authority researches such as literature reported in Nature Communications, because we have not much experience on metabolite extraction. We speculated this method is credible. We have added this paper as reference No. 27.
